# Information Design for Multiple Interdependent Defenders: Work Less, Pay Off More

Chenghan Zhou [1], Andrew Spivey [2], Haifeng Xu [3,*] and Thanh H. Nguyen [2]

1  Department of Computer Science, Princeton University, Princeton, NJ 08544, USA
2  Department of Computer and Information Science, University of Oregon, Eugene, OR 97403, USA
3   Department of Computer Science and Data Science Institute, University of Chicago, Chicago, IL 60605, USA
*  Correspondence: haifengxu@uchicago.edu

**Abstract:** This paper studies the problem of information design in a general security game setting in which multiple self-interested defenders attempt to provide protection simultaneously for the same set of important targets against an unknown attacker. A principal, who can be one of the defenders, has access to certain private information (i.e., attacker type), whereas other defenders do not. We investigate the question of how that principal, with additional private information, can influence the decisions of the defenders by partially and strategically revealing her information. In particular, we develop a polynomial time ellipsoid algorithm to compute an optimal private signaling scheme. Our key finding is that the separation oracle in the ellipsoid approach can be carefully reduced to bipartite matching. Furthermore, we introduce a compact representation of any ex ante persuasive signaling schemes by exploiting intrinsic security resource allocation structures, enabling us to compute an optimal scheme significantly faster. Our experiment results show that by strategically revealing private information, the principal can significantly enhance the protection effectiveness for the targets.

**Keywords:** information design; security games; persuasion

## 1. Introduction

In many real-world security domains, there are often multiple self-interested security teams who conduct patrols over the same set of important targets without coordinating with each other [1]. Among others, an important motivating domain of this paper is wildlife conservation: while patrol teams from various NGOs or provinces patrol within the same conservation area to protect wildlife from poaching, different NGOs or provinces typically have different types of targeted species and tend to operate separately (for instance, the Snow Leopard Foundation in Pakistan cares about leopards, whereas the Pakistan Bird Conservation Network primarily focuses on watching and protecting endangered birds [2]). Similarly, there are multiple different countries that simultaneously plan their own anti-crime actions against illegal fishing in international waters [3].

The study of multi-defender security games has attracted much recent attention. Unfortunately, most findings so far are relatively *negative*. Specifically, [4] showed that the lack of coordination among defenders may significantly lessen the overall protection effectiveness, leading to *unbounded* price of anarchy. In addition, [5] recently showed that finding a Nash Stackelberg equilibrium among the defenders, taking into account the strategic response of the attacker, is computationally *NP-hard*. Given these negative results, this paper asks the following question:

*How can one obtain defense effectiveness and computation efficiency in multi-defender security games?*

To answer the above question, we exploit the use of *information* as a "knob" to coordinate strategic agents' decisions. Specifically, we study how a principal with privileged private information (e.g., more accurate information about how much profit an attacker has

from poaching) can influence the decisions of all defenders by strategically revealing her information, a task also known as *information design* or *persuasion* [6]. Concretely, we study information design in a Bayesian security game setting with multiple self-interested defenders. These defenders attempt to protect important targets against an unknown attacker. The attacker *type* is unknown to the defenders. Nevertheless, all defenders share a common knowledge of a prior distribution over the attacker types. In this setting, there is a principal who has additional information about the attacker type and wants to communicate with both the defenders and the attacker through a persuasion signaling mechanism in order to influence all of their decisions towards the principal's goal.[1] In wildlife protection, for example, the principal may be the national park office, whereas different defenders correspond to different NGOs with their own focused species to protect. Since many poachers (or the attacker) are local villagers, park rangers can have access to private information through local informants about whom (i.e., which attacker type) is conducting poaching [9].

In summary, our results show that information design not only significantly improves protection effectiveness but also leads to efficient computation. Concretely, assuming the principal can communicate with defenders privately (also known as private signaling [10]), we develop an ellipsoid-based algorithm in which the separation oracle component can be decomposed into a polynomial number of sub-problems, and each sub-problem reduces to a bipartite matching problem. We remark that this by no means is an easy task, neither conceptually nor technically, since the outcomes of private signaling form the set of Bayes correlated equilibria [11], and computing an optimal correlated equilibrium is a fundamental and well-known intractable problem [12]. Our proof is technical and crucially explores the special structure of security games. In addition, we also investigate the ex ante private signaling scheme (a relaxation of private signaling in which the defenders and attacker decide whether to follow the principal's signals or not before any signal is realized [13]). In this scenario, we develop a novel compact representation for the principal's signaling schemes by compactly characterizing jointly feasible marginals. This finding enables us to significantly reduce the signaling scheme computation.

Finally, we present extensive experimental results evaluating our proposed algorithms in various game settings. We evaluate two different principal objectives: (i) maximizing the defenders' social welfare; and (ii) maximizing her own utility. Our results show that through signaling schemes, the principal can significantly increase the social welfare of the defenders while substantially reducing the attacker's utility.

*Comparison with Previous Works*

**Defense and Attack Models for Security:** There is extensive literature studying defense and attack models for security. Our discussion here cannot do justice to this large body of literature; we thus refer interested readers to an excellent survey paper [14], which classifies previously studied models according to three dimensions: system structure (eight types), defense measures (six types), and attack tactics and circumstances (seven types). Under their language, our work falls within the *multiple elements* system structure, *protection* defense measure, and *attack against single element* attack tactics. Prior to the writing of the survey article, not much previous work fell into this particular category. The most relevant publications for us are [15,16], which provide a systematic equilibrium analysis for the strategic game between a multi-resource defender and a single-element attacker. However, these works differ from ours in two key aspects: (1) their study is analytical, whereas our work is computational and tries to find the optimal defense strategy; (2) they consider simultaneous-move games, whereas our game is sequential and falls into the Stackelberg game framework. Specifically, we adopt the widely used class of games termed *Stackelberg Security games*, which capture strategic interactions between defenders and attackers in security domains [17]. This research advance comes slightly after the survey article [14], but has nevertheless led to a significant impact with deployed real-world applications, e.g., for airport security [18], ferry protection [19], and wildlife conservation [20].

Within the security game literature, most-relevant to our work is the recent study of multiple-defender security games. Several of them consider defenders to have identical interests [1,21] or to have their own disjointed sets of targets [4,22–25]. The game model in [5] is the most-related to ours. This previous work investigates the existence and computation of a Nash Stackelberg equilibrium among the defenders. To our knowledge, our work is the first to study information design in multi-defender security games. In contrast to previous negative results, our findings are much more encouraging. Our positive results even extend to more realistic game models with defender patrolling costs, which cannot be handled by the existing work.

**Strategic Information Disclosure for Security:** Information design, also known as signaling, has attracted much interest in various domains, such as public safety [7,8], wildlife conservation [26], traffic routing [13,27], and auctions [28,29]. Most-related to us is [30], which studies signaling in Bayesian Stackelberg games. All previous work assumes a single defender, whereas our paper tackles the complex *multiple-defender* setup. This requires us to work with exponentially large representations of signaling schemes and necessitates novel algorithmic techniques with compact representations.

**Other Learning-based Solutions:** Recent research in multi-agent reinforcement learning (MARL) has studied factors that influence agents' behavior in a shared environment. For example, [31] studies how to convey private information through actions in cooperative environments. Ref. [32] uses monetary reward (which they call 'causal inference reward') to influence opponents' actions. Unlike the tools studied in previous multi-agent reinforcement learning literature, our model takes advantage of information asymmetry between the principal and various stakeholders (including both defending agencies and the attacker) to influence their actions. Therefore, both our setup and approach are different from these previous learning-based methods.

## 2. Preliminary

We consider a general security game setting in which there are multiple self-interested defenders $\mathbf{D} = \{1, \ldots, |\mathbf{D}|\}$ who have to protect important targets $\mathbf{T} = \{1, \ldots, |\mathbf{T}|\}$ from an attacker. Each defender can protect at most one target.[2] The defenders do not know the attacker's type, but they share common prior knowledge about the distribution over possible attacker types: $\{q(\lambda)\}_{\lambda \in \Lambda}$ with $\Lambda = \{1, \ldots, |\Lambda|\}$, where $q(\lambda)$ is the probability that the attacker has type $\lambda$. If a defender $d$ decides to go to a target $t$, he has a patrolling cost of $C^d(t) < 0$. If the attacker $\lambda$ successfully attacks a target $t$, he receives a reward $R^\lambda(t) \geq 0$, while each defender $d$ receives a penalty $P^d(t) \leq 0$. Conversely, if any of the defenders catches the attacker $\lambda$ at $t$, the attacker receives a penalty $P^\lambda(t) < 0$, while each defender $d$ obtains a reward $R^d(t) > 0$. Notably, one defender suffices to fully protect a target, whereas multiple defenders on the same target are *not* any more effective. This leads to interdependence among defenders and is the major source of inefficiency without coordination [4].

## 3. Optimal Private Signaling

We first study the design of private signaling schemes that help the principal to coordinate the defenders. The principal leverages her private information about the attacker type to influence the decisions of all players (including the attacker) by strategically revealing her information. We adopt the standard assumption of information design [33] and assume that the principal commits to a signaling scheme $\omega$ and $\omega$ that is publicly known to all players. At a high level, a private signaling scheme generates a random variable called *signal profile* $\mathbf{s}$, which is correlated with $\lambda$, where $s(d)$ is the private signal sent to the defender $d$, and $s(a)$ is the signal sent to the attacker. Each defender $d$, once receiving a certain private signal $s_0$, updates his belief on the attacker type using Bayes rule as follows:

$$P(\lambda \mid s_0) = \frac{q(\lambda) \sum_{\mathbf{s}:s(d)=s_0} \omega(\mathbf{s} \mid \lambda)}{\sum_{\lambda'} q(\lambda') \sum_{\mathbf{s}:s(d)=s_0} \omega(\mathbf{s} \mid \lambda')}$$

where $\omega(\mathbf{s} \mid \lambda)$ is the probability the signal profile $\mathbf{s}$ is generated given the attacker type is $\lambda$.

Any private signaling scheme induces a Bayesian game among players. According to [11], all the Bayes–Nash equilibria that can possibly arise for any private signaling scheme form the set of *Bayes correlated equilibria* (BCEs). Similar to the standard correlated equilibria, the signals of a private signaling scheme in a BCE can also be interpreted as *obedient* action recommendations. Therefore, a private signal profile can be represented as $\mathbf{s} = (\{s(d)\}, s(a))$, where $s(d) \in \mathbf{T}$ is the suggested protection target for *defender* $d \in \mathbf{D}$, and $s(a) \in \mathbf{T}$ is the "suggested" target[3] for the attacker to attack. With slight abuse of notations, we use $s(-a)$ to represent the profile of signals sent to all the defenders, respectively, and $s(-a, -d)$ is the profile of signals sent to other defenders except defender $d$.

**Example 1.** *To illustrate the idea, let us consider a simple example motivated realistically by wildlife conservation. Suppose there are three main regions $A, B, C$ with important species, and there are two defending/patrolling agencies $d_1, d_2$. Defender $d_1$ (e.g., the Snow Leopard Foundation in Pakistan) only cares about species 1 (e.g., leopard), whereas defender $d_2$ only cares about species 2. Species 1 is distributed across only regions $A, B$, with more in A—specifically, defender $d_1$ suffers cost 9 (which is the poacher's gain) in region A but suffers cost 4 in region B for failing to protect them. species 2 is distributed mainly in targets $B, C$ but with more at C—specifically, defender $d_2$ suffers cost 4 at target B but cost 9 at target C. The defenders' utility for successfully protecting any target is normalized to 0 for ease of analysis. There are three possible types of poachers: those interested in poaching species 1, those interested in 2, and those interested in both. The two defending agencies share a common prior that a random poacher has type (1) or (2) each with probability 3/7 but has type (12) with probability 1/7 (i.e., fewer poachers are interested in both species).*

*We stand at the perspective of the national park office, which looks to optimize the protection of both species 1 and 2. We assume that the national park office can observe the exact type of the poacher,[4] and would like to strategically reveal this information to the two defending agencies in order to optimize the overall protection of the two species. For simplicity, suppose the national park does not have patrolling resources (though the illustrated idea would apply in that instance as well).*

*A natural first thought one may have is that the national park should be transparent and fully reveal the poacher type to both defenders. Unfortunately, this turns out not to be a good policy. If the poacher is type (12), then it is easy to verify that it is a unique Nash equilibrium for defender $d_1$ to patrol A with probability 3/4 and B otherwise, and $d_2$ to patrol C with probability 3/4 and B otherwise. This leads to overall protection probability $1 - (1 - \frac{1}{4})(1 - \frac{1}{4}) = 7/16$ at target B and 3/4 at A, C. The rational poacher type (12) will attack the more profitable target B, leading to cost $(4 + 4) \times (1 - 7/16) = 4.5$ to the national park. If the poacher has type (1) (or (2)), then only defender $d_1$ (or $d_2$) will conduct patrolling at target A, B. Simple calculation shows that the optimal $d_1$ strategy is to protect A with probability 9/13, leading to defender cost 36/13. This, in total, leads to expected defender cost $\frac{1}{7} \times (36/13) + 2 \times \frac{3}{7} \times (4.5) = \frac{29.77}{7}$.*

*It turns out that revealing different information privately to the two defenders leads to much higher utility. Specifically, consider the policy that reveals to defender $d_1$ whether the poacher is instead in species 2 (i.e., having type (2) or (12)) or not (i.e., having type (1)). In the first situations, $d_1$ knows that the poacher is definitely interested in species 2 but is unsure whether he is interested in 1—simple posterior updates shows that there is 1/4 probability that the poacher is interested in species 1 as well. So $d_1$ will nevertheless conduct patrolling just in case poacher is interested in species 1. Calculation shows that defender $d_1$'s equilibrium strategy will patrol target A with probability $p_1 = 0.65$ and B with 0.35. In the second situation, $d_1$ knows for sure that the poacher is interested in species 1, and his equilibrium strategy will now put probability $p_2 = 0.776$ on target A. Due to symmetry, defender $d_2$ has a symmetric strategy. Overall, it can be calculated that the defender will suffer cost $9 \times (1 - p_2) = 2.016$ whenever the poacher is interested in only one species and cost $p_1^2 \times (4 + 4) = 3.38$ when the poacher is interested in both. This overall leads to total defender cost $\frac{1}{7} \times (3.38) + 2 \times \frac{3}{7} \times (2.016) = \frac{15.476}{7}$. This reduces the cost $\frac{29.77}{7}$ under the transparent policy almost by half. Intuitively, this is because the private signaling policy carefully obfuscates the poacher type so that $d_2$ is incentivized to protect target B even when the poacher is not interested in species 2. This allows $d_1$ to focus more on the protection of A.*

### 3.1. An Exponential-Size LP Formulation

Like a typical formulation of optimal correlated equilibrium, optimal private signaling can also be formulated as an exponentially large linear program (LP). Specifically, the principal attempts to find an optimal signaling scheme $\Omega = \{\omega(\mathbf{s} \mid \lambda)\}$ to optimize her objective, which can be either her own utility (if she is a defender) or the social welfare of the defenders. We abstractly represent the principal's objective function with respect to a signal $\mathbf{s}$ as $U(\mathbf{s})$. The optimal private signaling can be formulated as the following LP:

$$\max \; \sum_{\lambda} q(\lambda) \sum_{\mathbf{s}} \omega(\mathbf{s} \mid \lambda) U(\mathbf{s}) \text{ s.t.} \tag{1}$$

(Attacker obedience) $\forall \lambda, t, t'$ :

$$\sum_{\mathbf{s}:t=s(a)} \omega(\mathbf{s} \mid \lambda) U^{\lambda}(\mathbf{s}) \geq \sum_{\mathbf{s}:t=s(a)} \omega(\mathbf{s} \mid \lambda) U^{\lambda}(s(-a), t') \tag{2}$$

(Defender obedience) $\forall d, t, t'$ :

$$\sum_{\lambda} q(\lambda) \sum_{\mathbf{s}:s(d)=t} \omega(\mathbf{s} \mid \lambda) U^{d}(\mathbf{s}) \tag{3}$$

$$\geq \sum_{\lambda} q(\lambda) \sum_{\mathbf{s}:s(d)=t} \omega(\mathbf{s} \mid \lambda) U^{d}(s(-a, -d), t', s(a))$$

$$\sum_{\mathbf{s}} \omega(\mathbf{s} \mid \lambda) = 1, \qquad \omega(\mathbf{s} \mid \lambda) \geq 0, \forall \mathbf{s}, \lambda \tag{4}$$

where (2) and (3) are obedience constraints that guarantee the attacker of any type and all defenders will follow the principal's recommendation. The utilities of each defender $d$ and each attacker type $\lambda$ are determined as follows:

$$U^{d}(\mathbf{s}) = C^{d}(s(d)) + P^{d}(s(a)), \text{ if } \forall d' : s(a) \neq s(d')$$
$$U^{d}(\mathbf{s}) = C^{d}(s(d)) + R^{d}(s(a)), \text{ if } \exists d' : s(a) = s(d')$$
$$U^{\lambda}(\mathbf{s}) = R^{\lambda}(s(a)), \text{ if } \forall d' : s(a) \neq s(d')$$
$$U^{\lambda}(\mathbf{s}) = P^{\lambda}(s(a)), \text{ if } \nexists d' : s(a) = s(d')$$

Problems (1)–(4) have an exponential number of variables $\{\omega(\mathbf{s} \mid \lambda)\}$ due to exponentially many possible defender allocations. This is also the common challenge in computing optimal correlated equilibria for succinctly represented games with many players (defenders in our case). Indeed, an optimal correlated equilibrium has been proven to be NP-hard in many succinct games [12]. Perhaps surprisingly, next, we show that LPs (1)–(4) can be solved in polynomial time in our case.

### 3.2. A Polynomial-Time Algorithm

We prove the following main positive result.

**Theorem 1.** *The optimal private signaling scheme can be computed in polynomial time.*

The rest of this section is devoted to the proof of Theorem 1. We elaborate the proof for the principal objective of maximizing the defender social welfare, i.e., $U(\mathbf{s}) = \sum_{d} U^{d}(\mathbf{s})$. The proof is similar when the principal is one of the defenders. Our proof is divided into three major steps and crucially exploits the structure of security games.

**Step 1: Restricting to simplified pure strategy space.** One challenge of designing the signaling scheme is when multiple defenders are recommended a same target, which significantly complicates computation of marginal target protection. Therefore, our first step is to simplify the pure strategy space to include only those in which all defenders cover different targets. To do so, we create **D** *dummy* targets for which rewards, penalties, and costs are zero for both the defenders and the attacker.[5] When the players choose one of these dummy targets, it means they choose to do nothing. As a result, we have $(\mathbf{T} + \mathbf{D})$ targets in total, including these dummy targets. The creation of these dummy targets does not influence the actual outcome of any signaling scheme but introduces a nice characteristic of

the optimal signaling scheme (Lemma 1). This characteristic of at most one defender at each target allows us to provide more efficient algorithms to find an optimal signaling scheme.

**Lemma 1.** *There is an optimal signaling scheme such that for any signal profile* **s** *with a positive probability (i.e., $\omega(\mathbf{s} \mid \lambda) > 0$), then $s(d) \neq s(d')$ for all $d \neq d'$.*

**Proof.** Let us assume that in a signaling scheme there is a signal for which multiple defenders are sent to the same target $t$. We revise this signal by only suggesting the defender $d$ with the lowest cost $C^d(t)$ to $t$ and other defenders are sent to dummy targets instead. First of all, the expected cost is reduced, while the coverage probability at each non-dummy target remains the same. As a result, the principal's objective does not change. Second, the attacker's obedience constraints do not change. Third, the LHS of the defender's obedience constraints increases, while the RHS is the same. This means no obedience constraint is violated. $\square$

**Step 2: Working in the dual space.** Since LP (1)–(4) has exponentially many variables, we first reduce it to the following dual linear program (1)–(4) via the standard linear duality theory [17], which turns out to be more tractable to work with:

$$\min \sum_\lambda \gamma(\lambda) \text{ s.t.} \tag{5}$$

$$\gamma(\lambda) + \sum_{t'} \Big[ U^\lambda(s(-a), t') - U^\lambda(\mathbf{s}) \Big] \alpha^\lambda(s(a), t') \tag{6}$$

$$+ q(\lambda) \sum_{d,t'} \Big[ U^d(s(-a, -d), t', s(a)) - U^d(\mathbf{s}) \Big] \beta^d(s(d), t')$$

$$\geq q(\lambda) U(\mathbf{s}), \forall(\mathbf{s}, \lambda)$$

$$\alpha^\lambda(t, t'), \beta^d(t, t') \geq 0, \forall \lambda, d, t, t'. \tag{7}$$

where each constraint in (6) corresponds to the primal variable $\omega(\mathbf{s} \mid \lambda)$. The dual variables $\alpha^\lambda(t, t')$ correspond to attacker obedience constraints (2). The dual variables $\beta^d(t, t')$ correspond to defender obedience constraints (3). Finally, the variables $\gamma(\lambda)$ correspond to constraints (4).

Problems (5)–(7) have an exponential number of constraints. We employ the ellipsoid method [35] by designing a polynomial–time separation oracle. In this oracle, given a value of $(\alpha^\lambda(t, t'), \beta^d(t, t'), \gamma(\lambda))$, it either establishes that this value is feasible for the problem or, if not, it outputs a hyper-plane separating this value from the feasible region. In the following, we focus on a particular type of oracle: those generating violated constraints. The oracle solves the following optimization problems; each corresponds to a fixed $\lambda$ and $s(a)$ (to be some target $t_0$),

$$\min_{\mathbf{s}: s(a) = t_0} \sum_{t'} \Big[ U^\lambda(s(-a), t') - U^\lambda(\mathbf{s}) \Big] \alpha^\lambda(t_0, t') \tag{8}$$

$$+ q(\lambda) \sum_{d,t'} \Big[ U^d(s(-a, -d), t', t_0) - U^d(\mathbf{s}) \Big] \beta^d(s(d), t')$$

$$- q(\lambda) U(\mathbf{s})$$

If the optimal objective of this problem is *strictly* less than $-\gamma(\lambda)$ for any $(\lambda, t_0)$, it means we found a violated constraint corresponding to $(\mathbf{s}^*, \lambda)$, where $\mathbf{s}^*$ is an optimal solution of (8). We iterate over every $(\lambda, t_0)$ to find all violated constraints and add them to the current constraint set.

**Step 3: Establishing an efficient separation oracle.** We now solve (8) for any given $(\lambda, t_0)$. We further divide this problem into two sub-problems; each can be solved via bipartite matching (which is polynomial time). More specifically, we divide the signal set $\{\mathbf{s} : s(a) = t_0\}$ into two different subsets, as elaborated in the following.

**Case 1 of Step 3:** Attacked target is not covered. The first subset consists of all signals such that $t_0 \notin s(-a)$; that is, none of the defenders are assigned to $t_0$. In this case, the

attacker will receive a reward $R^\lambda(t_0)$ for attacking $t_0$, while every defender $d$ receives a penalty $P^d(t_0)$. Thus, each of the following elements in (8) is straightforward to compute:

$$U^\lambda(s(-a), t') - U^\lambda(\mathbf{s}) = \begin{cases} P^\lambda(t') - R^\lambda(t_0) & \text{if } t' \in s(-a) \\ R^\lambda(t') - R^\lambda(t_0) & \text{if } t' \notin s(-a) \end{cases}$$

$$U^d(s(-a, -d), t', t_0) - U^d(\mathbf{s})$$
$$= \begin{cases} C^d(t') - C^d(s(d)) & \text{if } t' \neq t_0 \\ R^d(t_0) + C^d(t_0) - P^d(t_0) - C^d(s(d)) & \text{if } t' = t_0 \end{cases}$$

$$U(\mathbf{s}) = \sum_d P^d(t_0) + C^d(s(d))$$

Given the above computation, we observe that the second and third components (in the second and third lines) of the objective (8), which only depends on the defender utilities, consists of multiple terms—each term depends only on the allocation of each individual defender $(d, s(d))$. On the other hand, the first component (in the first line) of the objective, which depends on the attacker's utility, has terms which depend on targets not in the defender allocation. Therefore, in order to create a corresponding bipartite matching problem, we introduce $|\mathbf{T}|$ new dummy defenders and the following weights between $|\mathbf{T}| + |\mathbf{D}|$ defenders and $|\mathbf{T}| + |\mathbf{D}|$ targets:

$$\eta(d, t) = q(\lambda) \sum_{t' \neq t_0} \left[ C^d(t') - C^d(t) \right] \beta^d(t, t')$$
$$+ q(\lambda) \sum_{t' = t_0} \left[ R^d(t_0) + C^d(t_0) - P^d(t_0) - C^d(t)) \right] \beta^d(t, t')$$
$$+ [P^\lambda(t) - R^\lambda(t_0)] \alpha^\lambda(t_0, t) - q(\lambda) C^d(t), \forall t \neq t_0, d \leq |\mathbf{D}|$$
$$\eta(d, t_0) = +\infty, \forall d \leq |\mathbf{D}|$$
$$\eta(d, t) = [R^\lambda(t) - R^\lambda(t_0)] \alpha^\lambda(t_0, t), \forall t, \text{ dummy } d > |\mathbf{D}|$$

Weights associated with these dummy defenders correspond to the terms in (8), which depends on targets not in the actual defender allocation. The weight $\eta(d, t_0) = +\infty$ is to ensure that no actual defender in $\mathbf{D}$ will be assigned to $t_0$.

We now present Lemma 2 (which can be proved via a couple of algebraic computation steps), showing that Problem (8) becomes a Minimum Bipartite Matching between $|\mathbf{T}| + |\mathbf{D}|$ defenders and $|\mathbf{T}| + |\mathbf{D}|$ targets.

**Lemma 2.** *The problem (8) can now be reduced to the following bipartite matching problem using* $\eta(d, t)$:

$$\min_{\mathbf{m}} \sum_d \eta(d, m(d))$$

*after removing the constant term* $-q(\lambda) \sum_{d \in \mathbf{D}} P^d(t_0)$ *in (8). Here,* $m(d)$ *is a target matched to the defender $d$.*

**Case 2 of Step 3:** Attacked target is covered. On the other hand, the second subset consists of all signals such that $t_0$ is assigned to one of the defender. In this case, we further divide this sub-problem into multiple smaller problems by fixing the defender who covers $t_0$, denoted by $d_0$. Similar to *Sub-problem P1*, we introduce the following weights: $\forall t$

$$\eta(d, t) = q(\lambda) \sum_{t'} [C^d(t') - C^d(t)] \beta^d(t, t')$$
$$+ [P^\lambda(t) - P^\lambda(t_0)] \alpha^\lambda(t_0, t) - q(\lambda) C^d(t), \forall t, \forall d \in \mathbf{D} \setminus \{d_0\}$$
$$\eta(d, t) = [R^\lambda(t) - P^\lambda(t_0)] \alpha^\lambda(t_0, t), \forall t, \text{ dummy } d > |\mathbf{D}|$$

**Lemma 3.** *The problem ([8](#)) can now be reduced to the following bipartite matching problem using* $\eta(d,t)$:

$$\min_{\mathbf{m}} \sum_{d \neq d_0} \eta(d, m(d))$$

*after removing the constant terms* $\sum_{t' \neq t_0}[P^{d_0}(t_0) + C^{d_0}(t') - R^{d_0}(t_0) - C^{d_0}(t_0)]\beta^{d_0}(t_0, t') - q(\lambda)\sum_d R^d(t_0)$. *In addition,* $(d_0, t_0)$ *is removed from our matching setting.*

We now have the problem of a Minimum Bipartite Matching between $|\mathbf{T}| + |\mathbf{D}| - 1$ defenders to $|\mathbf{T}| + |\mathbf{D}| - 1$ targets, which can be solved in polynomial time.

## 4. Optimal Ex Ante Private Signaling

This section relaxes the private signaling requirement and assumes that players make decision on whether to follow signals or not before any signal is sent. Such ex ante private signaling has been studied recently in routing [13] and abstract games [36]. However, both works used the ellipsoid algorithm to compute the optimal scheme. While the ellipsoid algorithm is theoretically efficient, as we will show in our experiments, is it practically quite slow. In our case, we could have also just employed a similar technique. However, we take one step further and present a novel idea of using a compact representation of the signaling schemes such that the "reduced" signaling space becomes polynomial size for the number of targets. This important result helps to significantly scale up the problem computation.

### 4.1. An Exponential-Size LP Formulation

Overall, the problem of finding an optimal ex ante private signaling scheme can be formulated as the following LP, which has an exponential number of variables $\{\omega(\mathbf{s} \mid \lambda)\}$:

$$\max \sum_{\lambda} q(\lambda) \sum_{\mathbf{s}} \omega(s \mid \lambda)U(\mathbf{s}) \text{ s.t.} \tag{9}$$

(Attacker obedience) $\forall \lambda, t'$ :

$$\sum_s \omega(\mathbf{s} \mid \lambda)U^{\lambda}(\mathbf{s}) \geq \sum_{\mathbf{s}} \omega(\mathbf{s} \mid \lambda)U^{\lambda}(s(-a), t') \tag{10}$$

(Defender obedience) $\forall d, t'$ :

$$\sum_{\lambda} q(\lambda) \sum_{\mathbf{s}} \omega(\mathbf{s} \mid \lambda)U^d(\mathbf{s}) \tag{11}$$

$$\geq \sum_{\lambda} q(\lambda) \sum_{\mathbf{s}} \omega(\mathbf{s} \mid \lambda)U^d(s(-d), t', s(a))$$

$$\sum_{\mathbf{s}} \omega(\mathbf{s} \mid \lambda) = 1, \forall \lambda, \omega(\mathbf{s} \mid \lambda) \geq 0, \forall \mathbf{s}, \lambda \tag{12}$$

Similar to private signaling, we show that the optimal ex ante signaling scheme can be computed in polynomial time (Theorem [2](#)) by developing an ellipsoid algorithm.

**Theorem 2.** *The optimal private ex ante signaling scheme can be computed in polynomial time.*

### 4.2. Compact Signaling Representation

As we mentioned previously, while the ellipsoid algorithm is theoretically efficient, it runs slowly in practice. Therefore, we further show that in this scenario, we can provide a compact representation of signaling schemes such that the signaling space is polynomial for the number of targets. This immediately leads to a polynomial time algorithm for optimal ex ante private signaling by directly solving the polynomial-size linear program. Given any signaling scheme $\omega$, we introduce the new variable $\omega(a \to t, d \to t' \mid \lambda)$, which is the *marginal* probability that the attacker is sent to target $t$ and the defender $d$ is sent to target $t'$ given that the attacker type is $\lambda$. In addition, we introduce $\omega(a \to t \mid \lambda)$, which is the

probability the attacker is sent to $t$. Reformulating (9)–(11) based on these new variables is straightforward. For example, the objective (9) is reformulated as the following:

$$\sum_\lambda q(\lambda) \sum_{t,d} \omega(a \to t, d \to t \mid \lambda) R^d(t) + \sum_\lambda q(\lambda) \sum_t \left[ \omega(a \to t \mid \lambda) \right.$$
$$\left. - \sum_d \omega(a \to t, d \to t \mid \lambda) \right] \left[ \sum_d P^d(t) \right] + \sum_\lambda q(\lambda) \sum_{t',d} \left[ \sum_t \omega(a \to t, d \to t' \mid \lambda) \right] C^d(t')$$

The crux of this section is the following theorem. It fully characterize the conditions under which the compact representation corresponds to a feasible ex ante signaling scheme.

**Theorem 3.** *The following conditions are necessary and sufficient conditions to generate a feasible ex ante signaling scheme from a compact representation* $(\omega(a \to t \mid \lambda), \omega(a \to t, d \to t' \mid \lambda))$:

$$\sum_t \omega(a \to t \mid \lambda) = 1, \forall \lambda \tag{13}$$

$$\sum_{t'} \omega(a \to t, d \to t' \mid \lambda) = \omega(a \to t \mid \lambda), \forall \lambda, d \tag{14}$$

$$\sum_d \omega(a \to t, d \to t' \mid \lambda) \le \omega(a \to t \mid \lambda), \forall \lambda, t' \tag{15}$$

$$\omega(a \to t \mid \lambda) \ge 0, \omega(a \to t, d \to t' \mid \lambda) \ge 0, \forall \lambda, t, d, t' \tag{16}$$

**Proof.** It is obvious that these conditions are necessary. Let us consider $\{\omega(a \to t \mid \lambda)\}$ and $\omega(a \to t, d \to t' \mid \lambda)$ satisfying these conditions. We will show that these correspond to a feasible signaling scheme. First, we have:

$$\omega(d \to t' \mid a \to t, \lambda) = \frac{\omega(a \to t, d \to t' \mid \lambda)}{\omega(a \to t \mid \lambda)}$$

which is the probability of assigning defender $d$ to target $t'$ given the attacker is of type $\lambda$ and is assigned to target $t$. By fixing $\lambda$ and $a \to t_0$, we use $\omega(d \to t')$ as an abbreviation of $\omega(d \to t' \mid a \to t_0, \lambda)$ when the context is clear. We will prove that any $\{\omega(d \to t)\}$ satisfying the following conditions correspond to a feasible signaling scheme:

$$\sum_t \omega(d \to t) = 1, \forall d$$
$$\sum_d \omega(d \to t) \le 1, \forall t$$

In order to do so, we introduce the following general lemma:

**Lemma 4.** *For any a coverage vector* $\{\omega(d, t)\}$ *such that:*

$$\sum_t \omega(d, t) = r \tag{17}$$

$$\sum_d \omega(d, t) \le r, \tag{18}$$

*given* $0 \le r \le 1$, *there is an assignment of defenders to targets, denoted by* $(d_1, t_1), \ldots (d_{|\mathbf{D}|}, t_{|\mathbf{D}|})$, *such that:[6]*

- $\omega(d_i, t_i) > 0$ *for all* $i \in \{1, \ldots, |\mathbf{D}|\}$
- *Every maximally covered target* $t$, *i.e.,* $\sum_d \omega(d, t) = r$, *is assigned to a defender; that is,* $t \in \{t_1, \ldots, t_{|\mathbf{D}|}\}$.

**Proof.** Let $\mathbf{D}(t) = \{d : \omega(d, t) > 0\}$ be the support defender set of target $t$. Similarly, we also denote by $\mathbf{T}(d) = \{t : \omega(d, t) > 0\}$ the support target set of defender $d$. We divide the set of targets into two groups: (i) the group of all maximally covered targets $\mathbf{T}^{\text{high}} = \{t : \sum_d \omega(d, t)\} = r$; and (ii) the group of other targets $\mathbf{T}^{\text{low}} = \{t : \sum_d \omega(d, t) < r\}$. Without loss of generality, we represent $\mathbf{T}^{\text{high}} = \{t_1, \ldots, t_H\}$ and $\mathbf{T}^{\text{low}} = \{t_{H+1}, \ldots, t_{|\mathbf{T}|+|\mathbf{D}|}\}$, where $\{t_i\}$ is a permutation of targets $\{1, \ldots, |\mathbf{T}| + |\mathbf{D}|\}$.

**Step 1:** Inclusion of high-coverage target group $\mathbf{T}^{\text{high}}$. We first prove that there is a partial allocation from defenders to targets in $\mathbf{T}^{\text{high}}$, denoted by $(d_1, \ldots, d_H)$, such that

$d_i \in \mathbf{D}(t)$ for all $t_i \in \mathbf{T}^{\mathrm{high}}$, and they are pair-wise different, i.e., $d_i \neq d_j$ for all $t_i \neq t_j \in \mathbf{T}^{\mathrm{high}}$. We use induction with respect to $t$.

As a baseline, $t = 1$, the above statement holds true. Let us assume this statement is true for some $t < |\mathbf{T}^{\mathrm{high}}|$. We will prove that it is also true for $t + 1$. Let us denote by $(d_1, 1), \ldots, (d_t, t)$ the current sequence of defender-to-target assignments. At target $t + 1$, if there is $d \in \mathbf{D}(t + 1)$ such that $d \neq d_j$ for all $j \le t$, then we obtain a new satisfactory partial assignment $\{(d_1, 1), \ldots, (d_t, t), (d, t + 1)\}$.

Conversely, if $\mathbf{D}(t + 1) \subseteq \{d_1, \ldots, d_t\}$, without loss of generality, we assume $\mathbf{D}(t + 1) = \{d_1, d_2, \ldots, d_{t'}\}$ for some $t' \le t$. We obtain:

**Observation 1.** *There exists a target $t_0 \le t$ and a defender $d_0 \notin \{d_1, \ldots, d_t\}$ such that $\omega(d_0 \to t_0) > 0$.*

Indeed, if there is no such $(d_0, t_0)$, it means all targets $\{1, \ldots, t + 1\}$ can only be assigned to one of the defenders in $\{d_1, \ldots, d_t\}$. As a result, we will have:

$$r \times (t+1) = \sum_{j'=1}^{t+1} \sum_{d \in \mathbf{D}(j')} \omega(d, j') \le \sum_{j'=1}^{t} \sum_{j'' \in \mathbf{T}(d_{j'})} \omega(d_{j'}, j'')$$

$$= r \times t \text{ (contradiction)} \tag{19}$$

Now, if that target $t_0 \le t'$, then we obtain a new partial assignment $\{\ldots, (d_0, t_0), \ldots, (d_{t_0}, t + 1)\}$ by assigning $d_0$ to target $t_0$ and reallocating $d_{t_0}$ to $t + 1$ while keeping other assignments the same. On the other hand, if $t' < t_0 \le t$, it means $\mathbf{D}(j) \subseteq \{d_1, \ldots, d_t\}$ for all $j \le t'$. Without loss of generality, let us assume that target $t_0 = t' + 1$. We observe that there must exist a target $t_{00} \in \{1, \ldots, t\} \setminus \{t' + 1\}$ and a defender $d_{00} \notin \{d_1, \ldots, d_t\} \setminus \{d_{t'+1}\}$ such that $\omega(d_{00} \to t_{00}) > 0$. Indeed, if there is no such $(d_{00}, t_{00})$, it means all targets $\{1, \ldots, t + 1\} \setminus \{t' + 1\}$ can be only assigned to one of the defenders in $\{d_1, \ldots, d_t\} \setminus \{d_{t'+1}\}$. As a result, we have:

$$r \times t = \sum_{\substack{j'=1, j' \neq t'+1}}^{t+1} \sum_{d \in \mathbf{D}_{j'}} \omega(d, j')$$

$$\le \sum_{\substack{j'=1, j' \neq t'+1}}^{t} \sum_{j'' \in \mathbf{T}(d_{j'})} \omega(d_{j'}, j'')$$

$$= r \times (t-1) \text{ (contradiction)}$$

Now, if that target $t_{00} \le t'$ and $d_{00} = d_{t'+1}$, then we can do the swap $(d_{t'+1}, t_{00}), (d_{t_{00}}, t + 1), (d_0, t' + 1)$ while keeping other assignments the same. If that target $t_{00} \le t'$ and $d_{00} \neq d_{t'+1}$, then we can do a different swap $(d_{t_{00}}, t + 1), (d_{00}, t_{00})$. Finally, if $t_{00} > t' + 1$, without loss of generality, we assume $t_{00} = t' + 2$. We repeat the above analysis process until at some point we either find a feasible assignment or reach the following situation:

- $\exists d_0 \notin \{d_1, \ldots, d_t\}$ s.t $\omega(d_0 \to t' + 1) > 0$
- $\exists d_{00} \notin \{d_1, \ldots, d_t\} \setminus \{d_{t'+1}\}$ s.t. $\omega(d_{00} \to t' + 2) > 0$
- $\ldots$
- $\exists d_{\mathrm{final}} \notin \{d_1, \ldots, d_{t'}\}$ and $\exists t_{\mathrm{final}} \in \{1, \ldots, t'\}$ such that $\omega(d_{\mathrm{final}} \to t_{\mathrm{final}}) > 0$ where $\mathrm{final} = [0]^{t-t'}$.

In this situation, we first swap $(d_{t_{\mathrm{final}}}, t + 1), (d_{\mathrm{final}}, t_{\mathrm{final}})$. There are two cases. If $d_{\mathrm{final}} \notin \{d_1, \ldots, d_t\}$, then we found a solution. If $d_{\mathrm{final}}$ is equal to some $d_{t'+j}$ for some $j \le t - t'$, we then reassign $(d_{[0]^{t'+j}}, t' + j)$. At this step, there are two cases again. That is, either $d_{[0]^{t'+j}} \notin \{d_1, \ldots, d_t\}$, or $d_{[0]^{t'+j}}$ is one of $\{d_{t'+1}, \ldots, d_{t'+j-1}\}$. The former case means we found a solution, while the latter case indicates we have to do the reassignment again for a target in $\{t' + 1, \ldots, t' + j - 1\}$. Observe that every time we have to do a reassignment,

the index of the target for the reassignment is decreased. In the end, it will reach target $t' + 1$, for which we can reassign $d_0 \notin \{d_1, \ldots, d_t\}$ and obtain a feasible solution.

**Step 2:** Extension to include target group $\mathbf{T}^{\text{low}}$. We are going to prove that there is an assignment from defenders $\mathbf{D}$ to $|\mathbf{D}|$ targets that includes all targets in $\mathbf{T}^{\text{high}}$. We apply induction with respect to the defender $d$. Note that we cannot apply induction with respect to the targets $t$ since we include target group $\mathbf{T}^{\text{low}}$ in this analysis, and as a result, the equality on the LHS of (19) no longer holds.

As a baseline, we start with the feasible assignment of the group $\mathbf{T}^{\text{high}}$. Then at each induction step, we perform a defender–target swapping process that is similar to the case of the high-coverage target group $\mathbf{T}^{\text{high}}$. The tricky part is that for any swapping, we do not get rid of any targets that have been assigned so far (besides changing the defender assigned to them). This means that in the final assignment of the induction process, denoted by $(1, t_1), \ldots, (|\mathbf{D}|, t_{|\mathbf{D}|})$, all targets in $\mathbf{T}^{\text{high}}$ are still included. The details of this induction process are in the Appendix A.  □

Based on the result of Lemma 4, we allocate the following non-zero probability to the assignment with $r = 1$:

$$p = \min\{\min_d\{\omega(d, t_d)\}, r - \max_{t \notin \{t_1, \ldots, t_{\mathbf{D}}\}} \sum_d \omega(d, t)\}$$

Given this assignment, we update $w(d, t_d) = w(d, t_d) - p$ for all $d$. The resulting coverage vector $\{\omega(d, t)\}$ still satisfies the conditions (17) and (18) with the remaining $r = r - p < 1$. We keep doing this probability allocation until we obtain a feasible signaling scheme (i.e., $r$ reaches 0).  □

## 5. Experiments

In our experiments, we aim to evaluate both the solution quality and runtime performance of our algorithms in various game settings. All the LPs in our algorithms are solved with the CPLEX solver (version 20.1). We run our algorithms on a machine with an Intel i7-8550U CPU and 15.5 GB memory. The rewards and penalties of players are generated uniformly at random between $[0, 20]$ and $[-20, 0]$, respectively. All data points are averaged over 40 random games, and the error bars represents the standard error.

We compare our private and ex ante signaling schemes with: (i) a *baseline* method in which each defender optimizes his utility separately by solving a Bayesian Stackelberg equilibrium between that defender and the attacker without considering the strategies of the other defenders; and (ii) the Nash Stackelberg equilibrium (NSE) among the defenders. We use the method provided in [5] to approximate an NSE. We evaluate our signaling schemes in two scenarios corresponding to two different objectives of the principal: (i) maximizing the social welfare of the defenders (Figures 1–3); and (ii) maximizing her own defense utility (i.e., the principal is one of the self-interested defenders) (Figure 4). Next, we highlight our important results. Additional results can be found in the Appendix B.

In Figures 1 and 2, the x-axis is either the defender's cost range (the defense cost of each defender is randomly generated within this range), the number of targets, the number of defenders, or the number of attacker types. The y-axis is either the defender social welfare (Figure 1) or the average utility of the attacker (Figure 1). Note that in these figures, we do not consider the Nash Stackelberg equilibrium (NSE) among the defenders. This is because the method provided in [5] to approximate an NSE is only applicable for the no-patrolling-cost setting. Figure 1 shows that signaling schemes (`Private` and `ExAnte`) help to significantly increase the defender social welfare compared to the `Baseline` case. In addition, the defender social welfare in `ExAnte` is substantially higher than in the `Private` case. This result makes sense, since the persuasion constraints in `ExAnte` are less restricted. In addition, the social welfare is roughly a decreasing linear function of the cost range and the number of targets, while it increases linearly for the number of defenders. This is because the social welfare is a decreasing function of the defenders' coverage probability at

each target, and the higher the number of defenders is, the more coverage there is at each target. Conversely, we see an opposite trend in the attacker graphs (Figure 2).

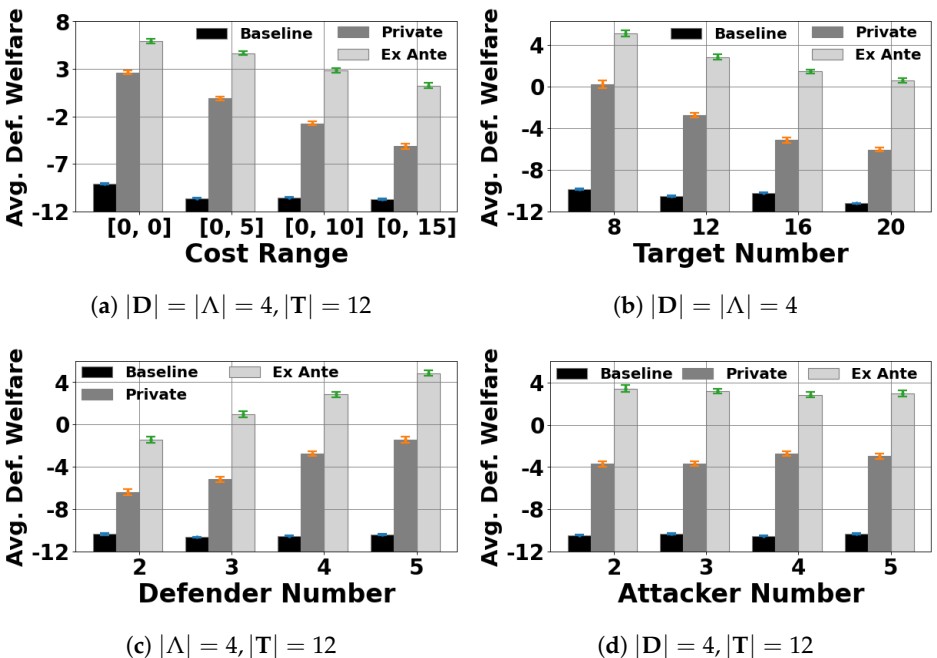

**Figure 1.** Average defender social welfare: the defenders' cost range is fixed to $[0, 10]$ in sub-figures (**b**–**d**).

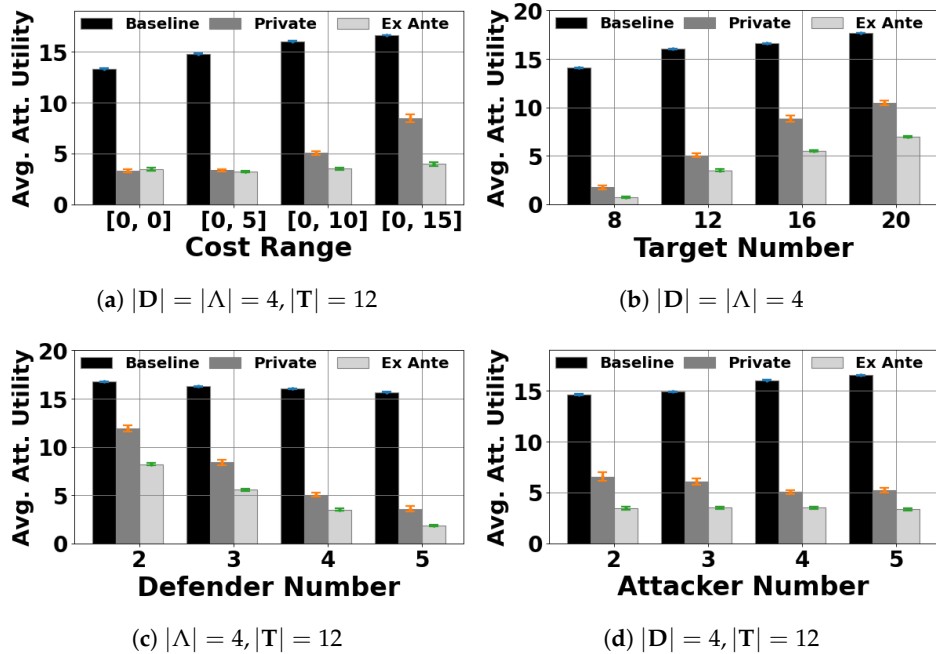

**Figure 2.** Average attacker utility: the defenders' cost range is fixed to $[0, 10]$ in sub-figures (**b**–**d**).

Furthermore, we include the NSE in our experiments with no defense cost. Figure 3 shows that despite `NashStackelberg` resulting in higher social welfare for the defenders compared to `Baseline`, in which each defender ignores the presence of other defenders, the social welfare in `NashStackelberg` is still significantly lower than that in `Private` and `ExAnte`. The results in Figures 1–3 clearly show that coordinating the defenders through the principal's signaling schemes helps to significantly enhance the protection effectiveness on the targets.

In Figure 4, we examine the situation in which the principal attempts to maximize her own utility (given she is one of the self-interested defenders). We again observe that the attacker suffers a significant loss in utility compared to `Baseline` (Figure 4a, `Private` and `ExAnte` versus `Baseline`). Conversely, the principal can get a significant benefit by strategically revealing her private information through the signaling mechanisms (Figure 4b).

Figure 5 shows the logarithm runtime of our algorithms compared to `Baseline` and `NSE`. We observe that our algorithms (`Private` and `ExAnte`) are suitable for medium games. In Figure 5a, it takes `Private` and `ExAnte` approximately 23 min and 40 s, respectively, to solve 20-target games. Furthermore, our compact representation method (`ExAnteCompact`) helps solve the signaling scheme significantly faster. It only takes `ExAnteCompact` approximately 2.7 s to solve 20-target games.

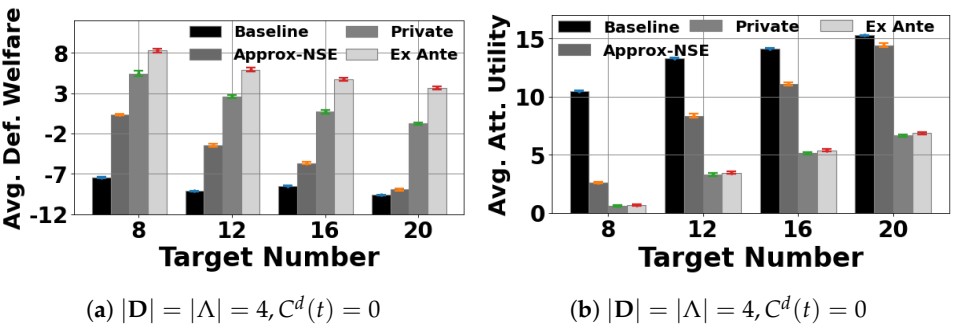

(**a**) $|\mathbf{D}| = |\Lambda| = 4, C^d(t) = 0$       (**b**) $|\mathbf{D}| = |\Lambda| = 4, C^d(t) = 0$

**Figure 3.** All evaluated algorithms, no patrolling costs.

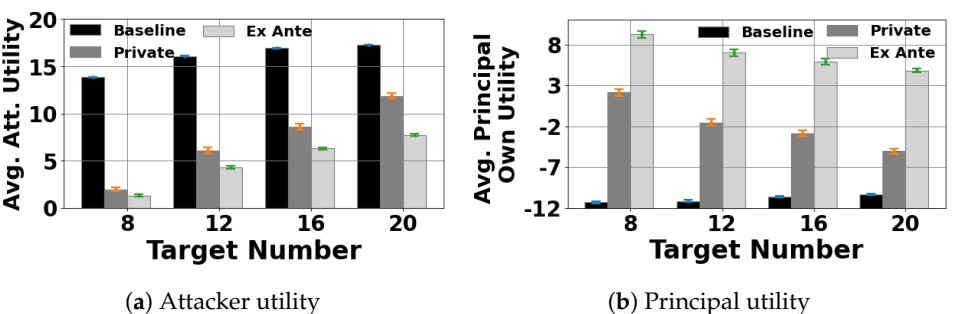

(**a**) Attacker utility       (**b**) Principal utility

**Figure 4.** The principal optimizes her own utility when $|\mathbf{D}| = |\Lambda| = 4$ and the defenders' cost range $C^d(t) \in [0, 10]$.

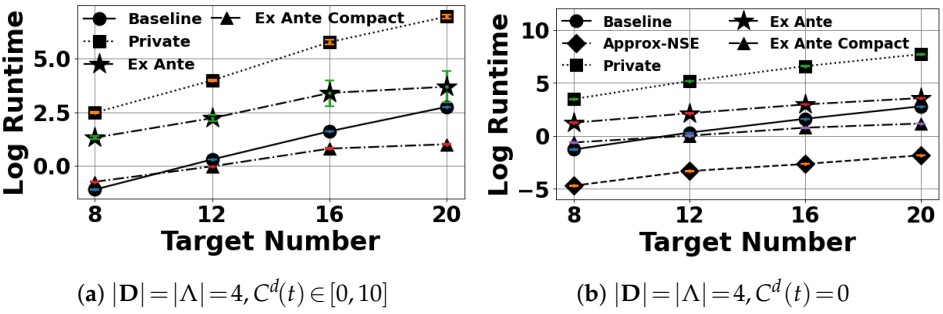

(**a**) $|\mathbf{D}| = |\Lambda| = 4, C^d(t) \in [0, 10]$       (**b**) $|\mathbf{D}| = |\Lambda| = 4, C^d(t) = 0$

**Figure 5.** Log run time in seconds.

Finally, we examine the performance in the ex ante case with a large number of targets or attacker types in Figure 6. Our algorithms can easily scale to about 160 targets, which is a large improvement compared to previous works [37,38]. We remark that it is typically impossible to test running time for such complicated security games for more than 200 targets on a single machine (most real-world applications such as conservation area

protection or border protection have fewer than 200 targets as well). For a large number of attacker types, our experiments show that the running time dependence of our algorithm with respect to the number of attacker types is linear, which is extremely efficient.

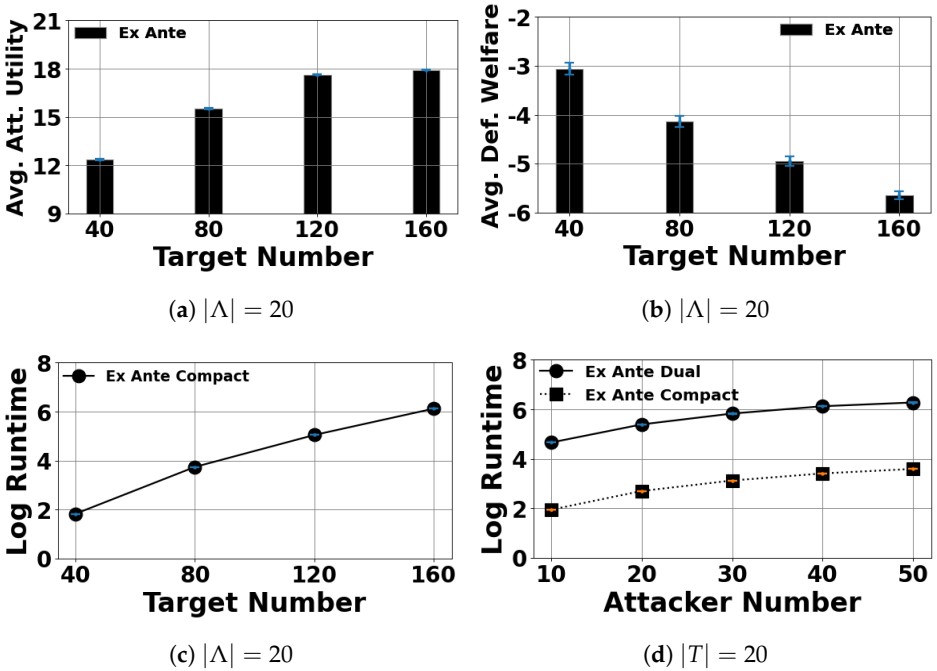

**Figure 6.** Scalability of target number or attacker types in ex ante setting when $|D| = 4$ and the defenders' cost range $C^d(t) \in [0, 10]$.

## 6. Summary

In this paper, we study information design in a Bayesian security game setting with multiple interdependent defenders. Our results (both theoretically and empirically) show that information design not only significantly improves protection effectiveness but also leads to efficient computation. In particular, in computing an optimal private signaling scheme, we develop an ellipsoid-based algorithm in which the separation oracle component can be decomposed into a polynomial number of sub-problems, and each sub-problem reduces to a bipartite matching problem. This is a non-trivial task, since the outcomes of private signaling form the set of Bayes correlated equilibria, and computing an optimal correlated equilibrium is a fundamental and well-known intractable problem. Our proof is technical and crucially explores the special structure of security games. Furthermore, we investigate the ex ante private signaling scheme. In this scenario, we develop a novel compact representation for the signaling schemes by compactly characterizing jointly feasible marginals. This finding enables us to significantly reduce the signaling scheme computation compared to the ellipsoid approach (which is efficient in theory but slow in practice).

**Author Contributions:** Conceptualization, H.X. and T.H.N.; Formal analysis, C.Z.; Funding acquisition H.X. and T.H.N.; Investigation, C.Z.; Methodology C.Z., H.X., and T.H.N.; Resources, C.Z.; Software, C.Z. and A.S.; Supervision H.X. and T.H.N.; Validation, C.Z.; Visualization, C.Z.; Writing—original draft, C.Z.; Writing—review and editing, C.Z., H.X., and T.H.N. All authors have read and agreed to the published version of the manuscript.

**Funding:** This research received no external funding.

**Data Availability Statement:** Not applicable.

**Acknowledgments:** Haifeng Xu is supported by an NSF grant CCF-2132506; this work is done while Xu is at the University of Virginia. Thanh H. Nguyen is supported by ARO grant W911NF-20-1-0344 from the US Army Research Office.

**Conflicts of Interest:** The authors declare no conflict of interest.

**Abbreviations**

The following abbreviations are used in this manuscript:

| | |
|---|---|
| BCE | Bayes correlated equilibrium |
| CPU | Central processing unit |
| LHS | Left-hand side |
| LP | Linear program |
| MARL | Multi-agent reinforcement learning |
| NSE | Nash Stackelberg equilibrium |
| RHS | Right-hand side |

**Appendix A. Proof of Lemma 4**

*Extension to Include Target Group* $\mathbf{T}^{\text{low}}$

We are going to prove that there is an assignment from defenders $\mathbf{D}$ to $|\mathbf{D}|$ targets that includes all targets in $\mathbf{T}^{\text{high}}$. We apply induction with respect to the defender $d$. Note that we cannot apply induction with respect to the targets $t$ since we include target group $\mathbf{T}^{\text{low}}$ in this analysis, and as a result, the equality on the LHS of (19) no longer holds.

As a baseline, without loss of generality, we have $(1, t_1), \ldots, (h, t_h)$ as the feasible assignment where $h = |\mathbf{T}^{\text{high}}|$ as the result of the group $\mathbf{T}^{\text{high}}$. Let us assume it is true for some $h \leq d \leq |\mathbf{D}|$. We will prove that it is true for $d + 1$. If there is a $t \in \mathbf{T}(d + 1)$ such that $t \notin \{t_1, \ldots, t_d\}$, then we obtain the new assignment with additional $(d + 1, t)$.

Conversely, if $\mathbf{T}(d + 1) \subseteq \{t_1, \ldots, t_d\}$. Let us assume $\mathbf{T}(d + 1) = \{t_1, \ldots, t_{d'}\}$ for some $d' \leq d$. We observe that there must exist a defender $d_0 \in \{1, \ldots, d\}$ and a target $t_0 \notin \{t_1, \ldots, t_d\}$ such that $\omega(d_0, t_0) > 0$. Indeed, if there is no such $(d_0, t_0)$, it means all defenders in $\{1, \ldots d + 1\}$ have to be assigned to targets in $\{t_1, \ldots, t_d\}$. As a result,

$$r \times (d + 1) = \sum_{d'=1}^{d+1} \sum_{t \in \mathbf{T}_d} \omega(d', t) \leq \sum_{j=1}^{d} \sum_{d \in \mathbf{D}_{t_j}} \omega(d, t_j) \leq r \times d.$$

which is contradictory.

Now if that defender $d_0 \leq d'$, then we obtain a new partial assignment by updating $(d_0, t_0), (d + 1, t_{d_0})$. On the other hand, let us consider the case when $d_0$ has to be outside of $\{1, \ldots, d'\}$. This means $\mathbf{T}(m) \subseteq \{t_1, \ldots, t_d\}$ for all $m \leq d'$. Without loss of generality, we assume $d_0 = d' + 1$. We apply the same procedure until we find a solution or reach the following situation:

- $\exists t_0 \notin \{t_1, \ldots, t_d\}$ s.t $\omega(d' + 1 \to t_0) > 0$
- $\exists t_{00} \notin \{t_1, \ldots, t_d\} \setminus \{t_{d'+1}\}$ s.t. $\omega(d_{00} \to d' + 2) > 0$
- $\ldots$
- $\exists t_{\text{final}} \notin \{t_1, \ldots, t_{d'}\}$ and $\exists d_{\text{final}} \in \{1, \ldots, d'\}$ such that $\omega(d_{\text{final}} \to t_{\text{final}}) > 0$ where final $= [0]^{d-d'}$.

We can do a similar swapping. Note that for any swapping, we do not get rid of any targets that have been assigned so far (besides changing the defender assigned to them). This means that in the final assignment, denoted by $(1, t_1), \ldots, (|\mathbf{D}|, t_{|\mathbf{D}|})$, all targets in $\mathbf{T}^{\text{high}}$ are included.

**Appendix B. Additional Experiments**

This section shows additional results varying the number of defenders and attackers.

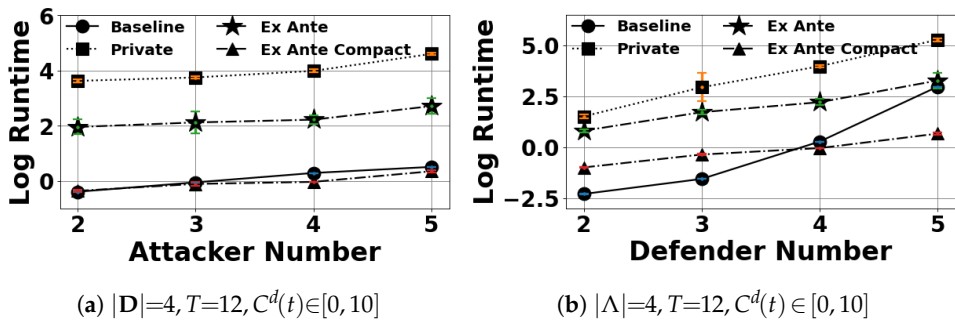

**(a)** $|\mathbf{D}|{=}4$, $T{=}12$, $C^d(t){\in}[0,10]$   **(b)** $|\Lambda|{=}4$, $T{=}12$, $C^d(t){\in}[0,10]$

**Figure A1.** Log runtime performance.

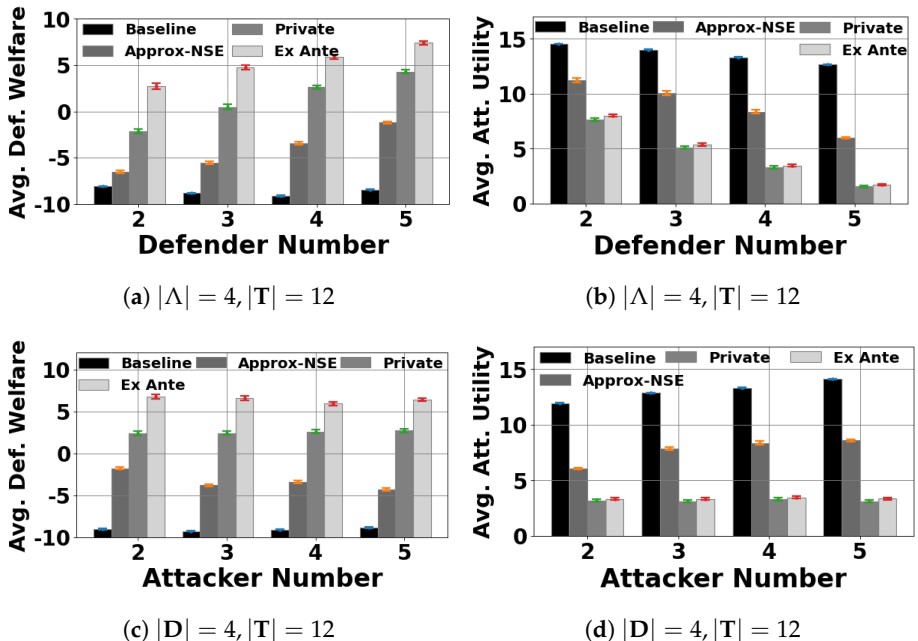

**(a)** $|\Lambda| = 4$, $|\mathbf{T}| = 12$   **(b)** $|\Lambda| = 4$, $|\mathbf{T}| = 12$

**(c)** $|\mathbf{D}| = 4$, $|\mathbf{T}| = 12$   **(d)** $|\mathbf{D}| = 4$, $|\mathbf{T}| = 12$

**Figure A2.** All evaluated algorithms, no patrolling costs.

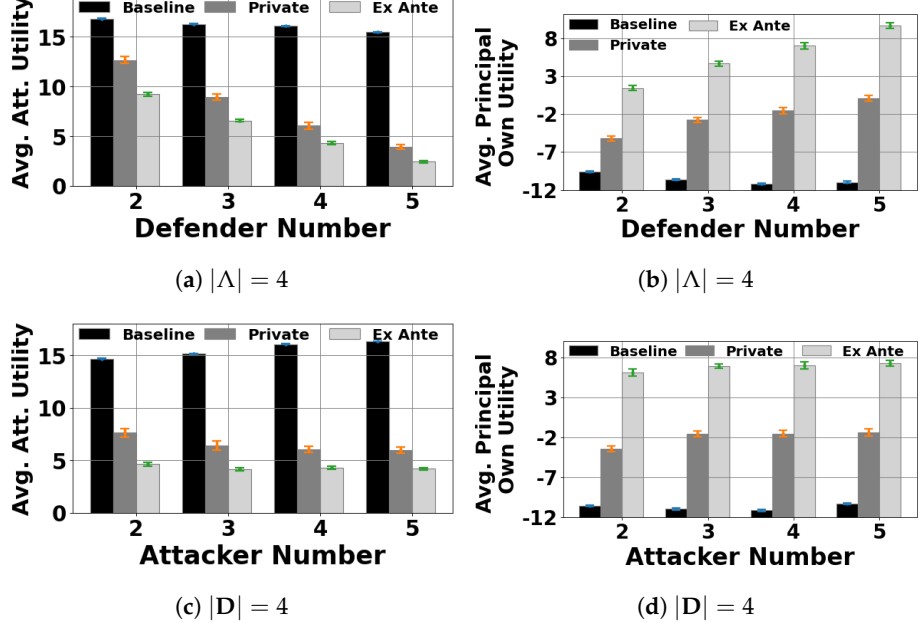

**(a)** $|\Lambda| = 4$   **(b)** $|\Lambda| = 4$

**(c)** $|\mathbf{D}| = 4$   **(d)** $|\mathbf{D}| = 4$

**Figure A3.** The principal optimizes her own utility when $|\mathbf{T}| = 12$ and the defenders' cost range $C^d(t){\in}[0,10]$.

## Notes

1    Notably, the defender can signal to the attacker as well, to either deter him from attacking or induce him to attack a specific target in order to catch him. Previous works have shown that this can benefit the defender [7,8] even though the attacker is fully aware of the strategic nature of the signal and will best respond to the revealed information.

2    This is without loss of generality, since any defender who can cover multiple targets can be "split" into multiple defenders with the same utilities.

3    The term "suggested" here should only be interpreted mathematically—i.e., given all the attacker's available information, $s(a)$ is identified as the *most profitable* target for the attacker to attack—and should not be interpreted as a real practice that the defender suggests the attacker to attack some target. Such a formulation, analogous to the revelation principle, is used for the convenience of formulating the optimization problem.

4    Such information can often by learned from informants such as local villagers [34].

5    In reality, such a dummy target could be unimportant infrastructure (e.g., a nearby rest area at the border of a national park with no animals around, as in wildlife conservation), which does not matter to any defender nor the attacker.

6    Since we have $(\mathbf{T} + \mathbf{D})$ targets in total while there are only $|\mathbf{D}|$ defenders, some targets will not be assigned to any defenders.

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
