# Peer review of "Information Design for Multiple Interdependent Defenders: Work Less, Pay Off More"

_games, doi:10.3390/g14010012_

Round 1

Reviewer 1 Report

I read a very interesting research paper. Congratulations for the hard work! Some comments:

Page 1, lines 33-34: What is a privileged private information? Attacker’s profile? How does the profile influence the choice of a target? How can one know before revealing the information that the information she detains is only in her hands, since all defenders are self-interested and know the signaling scheme? You need to clarify.

Page 2, line 42: Does the principal send a signal to the attacker to deter him from attacking or to induce him to attack a specific target where she can catch her?

Page 2, line 43: Since they all try to protect the same set of targets, in what do the goals differ from one defender to another? Are you minimizing the distance to the target?

Page 2, lines 63-64: If the defenders are self-interested, why would the principal maximize the defenders’ social welfare?

Page 3, line 91: What is MARL? Multi-agent reinforcement learning? Please write it entirely.

Page 3, line 92: information asymmetry between which parties?

Page 3, lines 119-120: s(-a): defenders? Not clear. Please reformulate.

Page 4, lines 139-144: Can you provide a real-life example of what a dummy target could mean?

Page 5, lines 161-163: Not clear. Please reformulate.

Page 6: What happens if d >= |D|?

Author Response

Thank you very much for the detailed review. We have addressed all your comments. In the following we explain the major changes we made based on your suggestions. 

1. Page 1, lines 33-34: What is a privileged private information? Attacker’s profile? How does the profile influence the choice of a target? How can one know before revealing the information that the information she detains is only in her hands, since all defenders are self-interested and know the signaling scheme? You need to clarify.

We have now provided more explanations on this matter. For instance, the private information may be the accurate information about how much profit an attacker has get from poaching, which is known to World Wildlife Fund (the coordinator) but may not be known by the patrollers at different provinces.   

Page 2, line 42: Does the principal send a signal to the attacker to deter him from attacking or to induce him to attack a specific target where she can catch her?

We added a footnote 1 to explain this. In a nutshell, both can happy, and the optimization program will automatically decide what option to choose based on the utility tradeoff and protection probabilities at each target. This is studied in much detailed by Reference [7] 

Page 2, line 43: Since they all try to protect the same set of targets, in what do the goals differ from one defender to another? Are you minimizing the distance to the target?

As we explained in the introduction, even though each defender protects the same set of targets, their utilities at different targets are different which make the strategies different. For instance, in Pakistan, multiple NGOs exist but different NGOs care about the protection of different species (e.g., Snow Leopard Foundation cares about Leopard whereas Pakistan Bird Conservation Network primarily focus on the watch and protection of endangered birds). So their protections strategies differ according to where their focused animals are more likely to show up, though their patrolling also help to protect animals at the same region.     

Page 2, lines 63-64: If the defenders are self-interested, why would the principal maximize the defenders’ social welfare?

This is a great question. Welfare is the most widely studied objective in previous works about multiple defender. However, our algorithmic approach should work for any principal's objective --- regardless it is his own or the sum of all defenders' utilities. 

Quick Search SI Title: Name:
Email: Conference: Add Reviewers [for GL] Submit

Reviewer 2 Report

The paper provides an efficient algorithm o solve a problem of a principal. There are attackers and defenders in the model. The principal sends signal to boh sides and seeks to maximize some abstract utility function.

The motivation is not completely clear for me. Why should the principal send signals to all sides, and not to defenders only? Why should attackers obey the principle? Why the pricnciple should have another utility function than defenders?

Also from prelimenaries section (section 2) it is not clear what is the meaning/interpretation of attackers' types.

I understand that the authors wish to provide a general abstract framework. But I suggest so include a realistic special case, with realistic utilities of and players, and maybe to provide an equilibrium outcome  in this special case, and to explain why this outcome is intuitive. This will clearify to readers a purpose of the whole exersice.

Author Response

We thank the reviewer for the careful review. The following are our response to the reviewer's major comments. These responses have also been integrated into our revised draft. 

Why should the principal send signals to all sides, and not to defenders only? Why should attackers obey the principle? Why the pricnciple should have another utility function than defenders?

We have refined our introduction section to explain these points. The reason that the principal send signals to the attacker is simply because this can further increase utility (see footnote 1 and references [7,7]), though it is certainly okay if the principal does not want to send signals to the attacker and is willing to settle at a less profitable strategy. In our model, we assumed the principal will try to maximize her utility. 

It is the attacker's best interest to obey the signals since the signals are designed to be "incentive compatible" (in information design, this is more often referred to as "obedient"). This is in spirit the same as in standard Stackelberg games, in which case the follower is predicted to take the best response action to any leader strategy. 

Also from prelimenaries section (section 2) it is not clear what is the meaning/interpretation of attackers' types.

Type here is the standard concept in Bayesian game theory, which is to describe any payoff-relevant information about the attacker/agent. Such information is unknown to the principal who however has some prior knowledge about it, captured by her prior distribution over types. 

Quick Search SI Title: Name:
Email: Conference: Add Reviewers [for GL] Submit

Round 2

Reviewer 1 Report

Thank you for the updates.

Author Response

Thank you very much for your comments, which helped a lot to improve the paper!

Reviewer 2 Report

I am satisfied with changes.

Author Response

(The authors gave the same response as above.)
